# High-Grade Surface Osteosarcoma of the Rib Mimicking a Neurogenic Tumor: Radiologic and Pathologic Findings

**DOI:** 10.3390/diagnostics13182880

**Published:** 2023-09-08

**Authors:** Kyungsoo Bae, Jin Il Moon, Hyo Jung An, Jae Jun Jung, Kyung Nyeo Jeon

**Affiliations:** 1Department of Radiology, Institute of Medical Science, Gyeongsang National University School of Medicine, Jinju 52727, Republic of Korea; ksbae@gnu.ac.kr (K.B.); logan63@gnu.ac.kr (J.I.M.); 2Department of Radiology, Gyeongsang National University Changwon Hospital, Changwon 51472, Republic of Korea; 3Department of Pathology, Gyeongsang National University School of Medicine, Jinju 52727, Republic of Korea; ari-el2020@naver.com; 4Department of Pathology, Gyeongsang National University Changwon Hospital, Changwon 51472, Republic of Korea; 5Department of Thoracic Surgery, Gyeongsang National University School of Medicine, Jinju 52727, Republic of Korea; thoracoscope@naver.com; 6Department of Thoracic Surgery, Gyeongsang National University Changwon Hospital, Changwon 51472, Republic of Korea

**Keywords:** rib, chest wall, osteosarcoma, surface osteosarcoma, computed tomography

## Abstract

Osteosarcoma commonly occurs in the intramedullary cavity of long bones such as the femur, tibia, and humerus in children and adolescents. Osteosarcoma occurring as a primary tumor in the chest wall is rare. Only a limited number of such cases have been documented in the existing literature. Herein, we present radiologic and pathologic findings of a high-grade surface osteosarcoma of the rib mimicking a neurogenic tumor in a 44-year-old woman.

A 44-year-old woman visited an outpatient clinic due to a nodular lesion detected on a chest X-ray screening. She did not complain about any symptoms. She was never a smoker. She was previously healthy. She had no notable family history of cancer. According to her chest CT scan, the nodule was an extrapleural lesion located along the inner surface of left eighth posterior rib. It measured about 1.8 cm in its maximum diameter. The nodule showed homogeneous contrast enhancement with a smooth well-demarcated margin. Focal cortical erosion of the left eighth rib was noted. Given the imaging features and the location of the lesion, a neurogenic tumor such as schwannoma was suspected. Nevertheless, upon closer examination of a magnified CT image using a bone window setting, the presence of bony spicules oriented perpendicularly to the rib cortex was observed, indicating a potential sunburst-like periosteal reaction. This finding raises concerns about the likelihood of malignancy (Figure 1).

We performed a video-assisted thoracoscopic excision of the nodule, along with adjacent ribs, for diagnostic and therapeutic purposes. Upon gross examination of the specimen, it was noted that the lesion abutted the eighth rib, with a very small portion of the nodule connected to the rib bone. Upon microscopic examination, the nodule was identified as a highly cellular spindle cell tumor. In a part of the lesion, woven bone with endochondral ossification was noted, suggestive of a high-grade osteosarcoma. A tiny part of the tumor was connected to the medullary cavity of the rib bone, indicating the potential presence of endosteal extension. Upon higher magnification, it was evident the malignant cells were infiltrating the medullary trabeculae (Figure 2). Based on pathologic findings, a diagnosis of high-grade surface osteosarcoma was established.

Osteosarcoma was reported to primarily involve the rib in only 1.3% of cases [1]. Unlike typical osteosarcomas that originate within the medullary cavity, juxtacortical osteosarcomas that constitute 4–10% of the entire spectrum of osteosarcomas have their origin on the outer surface of the bone cortex [2,3]. Based on their point of origin, juxtacortical osteosarcomas can be subdivided into three types, parosteal, periosteal, and high-grade surface types, corresponding to the external periosteal layer, internal periosteal layer, and any location within the periosteum, respectively. Such subtypes exhibit disparities not just in radiological and pathological characteristics but also in treatment strategies and eventual prognoses [3]. Of these subtypes, high-grade surface osteosarcomas carry the most unfavorable prognosis. The treatment protocol involves initiating neoadjuvant chemotherapy followed by an extensive surgical resection.

We presented the radiologic and pathologic findings of a high-grade surface osteosarcoma that emerged in the intercostal space. Despite its resemblance to a neurogenic tumor, the detection of a discreet periosteal reaction in the adjacent rib cortex on CT images prompted the consideration of a potential malignancy.

## Figures and Tables

**Figure 1 diagnostics-13-02880-f001:**
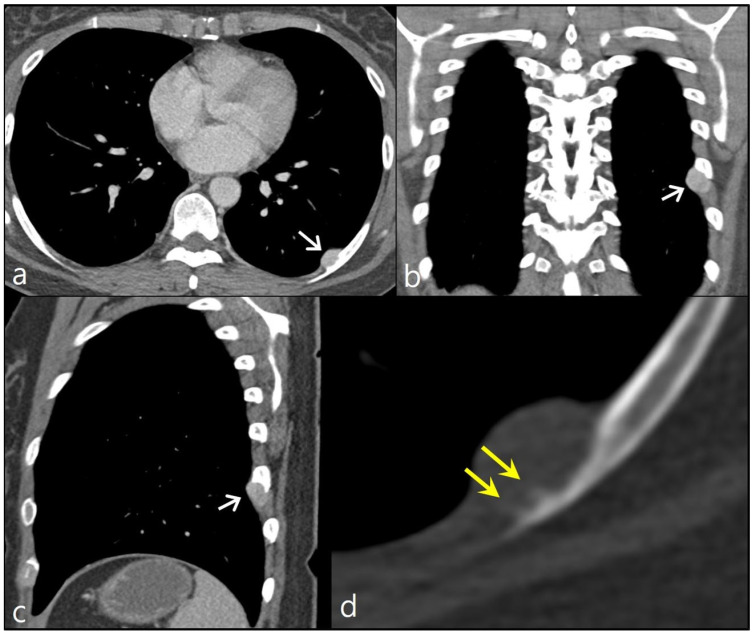
Chest CT images with contrast enhancement show a well-circumscribed nodule along the inner and inferior aspects of the left eighth posterior rib (arrows in **a**–**c**). Focal cortical erosion of the rib is noted. On the magnified image using a bone window setting, bony spicules (double arrow) oriented perpendicularly to the rib cortex were observed, indicating a potential sunburst-like periosteal reaction (**d**).

**Figure 2 diagnostics-13-02880-f002:**
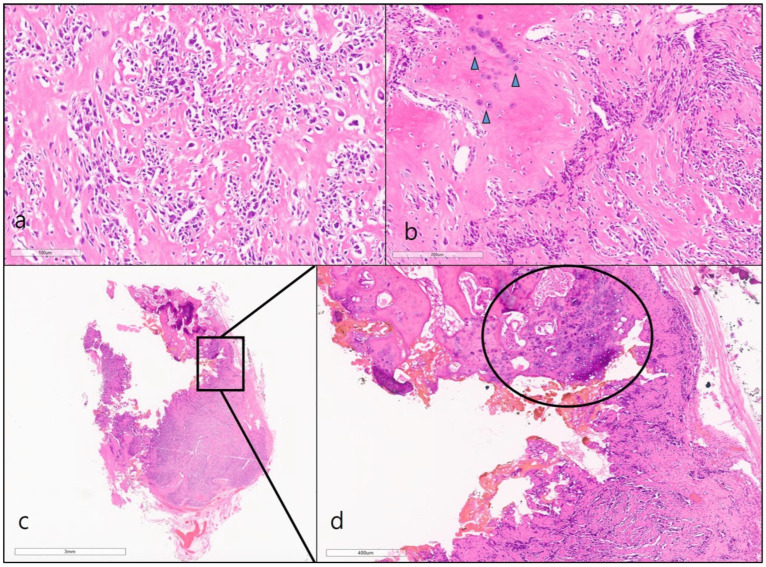
(**a**) A high cellular spindle cell tumor was composed of pleomorphic hyperchromatic cells with lace-like woven bones (×200, H&E). (**b**) In some parts of the lesion, there was woven bone with endochondral ossification (arrowheads), suggestive of a high-grade osteosarcoma (×100, H&E). A minute segment of the tumorous lesion displayed a connection to the medullary cavity of the rib bone, suggestive of endosteal extension (highlighted in box in (**c**), ×10, H&E). Upon closer magnification, it was evident that cancerous cells were infiltrating into medullary trabeculae (encircled in (**d**), ×100, H&E).

## Data Availability

Data are contained within the article. No new data were created or analyzed in this study.

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
