# Peer review of "High-Grade Surface Osteosarcoma of the Rib Mimicking a Neurogenic Tumor: Radiologic and Pathologic Findings"

_diagnostics, 2023, doi:10.3390/diagnostics13182880_

Round 1

Reviewer 1 Report

Thank you for bringing up this case to the scientific community. It is scientifically important to mention any cases of osteosarcoma, as there are elevated diagnosed cases worldwide, especially the cases that show no significant symptoms like the current manuscript. This case is extremely rare pathology, and it is interesting for the scientific community to know about it. The data results are so clear and conclusive. It could be great if you could mention more information about the patient background history such as if anyone in her family has diagnosed with cancer particularly osteosarcoma. Also, I am curious if the patient sample has been sequenced? Is there any factual reason for considering this case as a neurogenic tumor?

Author Response

(Answers to reviewer’s comments)

Reviewer 1

Thank you for bringing up this case to the scientific community. It is scientifically important to mention any cases of osteosarcoma, as there are elevated diagnosed cases worldwide, especially the cases that show no significant symptoms like the current manuscript. This case is extremely rare pathology, and it is interesting for the scientific community to know about it. The data results are so clear and conclusive. It could be great if you could mention more information about the patient background history such as if anyone in her family has diagnosed with cancer particularly osteosarcoma. Also, I am curious if the patient sample has been sequenced?

Answer) We thank the reviewer for pointing this out. The patient did not have any specific family history of cancer including osteosarcoma. The patient's sample has not undergone sequencing.

Is there any factual reason for considering this case as a neurogenic tumor?

Answer) The patient was asymptomatic. The tumor was predominantly located extraosseously, specifically on the inner surface of the left 8th rib, and showed no obvious mineralization. We interpreted the bone alteration as simple erosion rather than indicative of speculation (periosteal reaction). Therefore, we initially concluded that it was a peripheral nerve sheath tumor originating from the intercostal nerve, which is one of the most common chest wall tumors.

We thank the reviewer for his/her valuable comments and suggestions. These comments and suggestions have improved the quality of our manuscript.

Reviewer 2 Report

Dear Authors,

Thank you very much for the opportunity to review this case report.

It highlights the importance of tumor margins when assessing bone lesions.

To better state the point, a little paragraph explaining other tumoral margin patterns ( for example moth eaten, sclerotic...) could be useful.

Furthermore, this patient went right to excision without FNA/CNB. I do agree with this decision, but would be better to detail why this step was skipped.

Author Response

(Answers to reviewer’s comments)

Reviewer 2

Thank you very much for the opportunity to review this case report.

It highlights the importance of tumor margins when assessing bone lesions.

To better state the point, a little paragraph explaining other tumoral margin patterns (for example moth eaten, sclerotic...) could be useful.

Answer) We thank the reviewer for pointing this out. We have added a description explaining tumor margin and adjacent bone change as follows: 

  • The nodule showed homogeneous contrast enhancement with smooth well demarcated margin.
  • the presence of bony spicules oriented perpendicularly to the rib cortex was observed, indicating a potential sunburst-like periosteal reaction.

Furthermore, this patient went right to excision without FNA/CNB. I do agree with this decision, but would be better to detail why this step was skipped.

Answer) I Fine-needle aspiration or core biopsy for chest wall masses is not a routine procedure in our hospital, except in cases where the tumor is substantial in size or there is a significant suspicion of myeloma or metastatic disease. We conducted a video-assisted thoracoscopic excision of the nodule, along with adjacent ribs, for diagnostic and therapeutic purposes.

We thank the reviewer for his/her valuable comments and suggestions. These comments and suggestions have improved the quality of our manuscript.
